# Economic cost of patients with trisomy 13, 18, and 21 in a tertiary hospital in Thailand

**Preechaya Wongkrajang**[1,2], **Jiraphun Jittikoon**[3], **Wanvisa Udomsinprasert**[3], **Pattarawalai Talungchit**[4,5], **Usa Chaikledkaew**[5,6]*

1 Social, Economic and Administrative Pharmacy (SEAP) Graduate Program, Faculty of Pharmacy, Mahidol University, Bangkok, Thailand, 2 Department of Clinical Pathology, Faculty of Medicine Siriraj Hospital, Mahidol University, Bangkok, Thailand, 3 Department of Biochemistry, Faculty of Pharmacy, Mahidol University, Bangkok, Thailand, 4 Department of Obstetrics and Gynecology, Faculty of Medicine Siriraj Hospital, Mahidol University, Bangkok, Thailand, 5 Mahidol University Health Technology Assessment (MUHTA) Graduate Program, Mahidol University, Bangkok, Thailand, 6 Social and Administrative Pharmacy Division, Department of Pharmacy, Faculty of Pharmacy, Mahidol University, Bangkok, Thailand

* usa.chi@mahidol.ac.th

**Data Availability Statement:** Data cannot be shared publicly because our study obtained data from patients with trisomy 13, 18, and 21 which contained sensitive patient information and there

## Abstract

The purpose of this study was to determine direct and indirect costs of patients with trisomy (T) 13, 18, and 21 in Thailand. Direct medical costs were obtained from Siriraj Informatics and Data Innovation Center (SiData+), Faculty of Medicine, Siriraj Hospital, and indirect costs were estimated using a human capital approach. About 241 patients with T21 had outpatient care visits and 124 patients received inpatient care. For T13 and T18, five and seven patients were analyzed for outpatient and inpatient cares, respectively. For patients with T13, T18, and T21 receiving outpatient care, total annual mean direct medical costs ranged from 183.2 USD to 655.2 USD. For inpatient care, average yearly direct medical costs varied between 2,507 USD to 14,790 USD. The mean and median increased with age. In outpatient care, costs associated with drugs and medical devices were a major factor for both T13 and T21 patients, whereas laboratory costs were substantial for T18 patients. For inpatient care, costs of drug and medical devices were the greatest for T13 patients, while service fee and operation costs were the highest for T18 and T21 patients, respectively. For outpatient care, adult patients with congenital heart disease (CHD) had significantly higher mean annual direct medical costs than those without CHD. However, all adult and pediatric patients with CHD receiving inpatient care had significantly higher costs. Patients with T13, T18, and T21 had relative lifetime costs of 22,715 USD, 11,924 USD, and 1,022,830 USD, respectively.

## Introduction

Aneuploidies or chromosomal abnormalities affecting 0.5 to 1.0% of live births may cause a variety of health problems, with trisomy 21, 18, and 13 being the most prevalent [1]. One in every 700–800 live births is affected with trisomy 21 (T21) or Down syndrome, and the average life expectancy of those affected has grown by 50 years in the last century [2]. There are several clinical issues with varying degrees of severity, including poor muscle tone, intellectual and developmental disabilities, heart disease, gastrointestinal defect, hematologic abnormalities,

are ethical restrictions on publicly sharing a sensitive data set. Data are available from the Human Research Protection Unit, Faculty of Medicine Siriraj Hospital, Mahidol University (contact via Room 210, 2nd floor, His Majesty the King's 80th Birthday Anniversary 5th December 2007 Building, 2 Wang Lang Road Bangkoknoi, Bangkok 10700 or siethics@mahidol.ac.th) for researchers who meet the criteria for access to confidential data.

**Funding:** This study receives funding support from the Health Systems Research Institute (HSRI). The funders had no role in study design, data collection and analysis, decision to publish, or preparation of the manuscript.

**Competing interests:** The authors have declared that no competing interests exist.

hypothyroidism, and other endocrine abnormalities [3, 4]. Trisomy 18 (T18), often known as Edwards syndrome, is the second most frequent kind of trisomy. The incidence of T18 is one in 3,000 to 8,000 live births and the ratio of females to males is 3:1. More than 95% of these fetuses are lost due to spontaneous abortion. Less than 1% of affected infants survive until their first birthday [1, 5]. Most affected patients have a particular phenotype, including craniofacial features such as dolichocephaly with a prominent occiput, and clinical abnormalities such as heart disease, ophthalmologic and otolaryngologic, musculoskeletal, etc. It has been reported that girls with T18 have a greater life expectancy than boys [6–9]. Trisomy 13 (T13) or Patau syndrome is the third most common trisomy and the incidence of T13 is 1 in 12,000 to 16,000 live births. Over 95% of these fetuses are lost due to spontaneous abortion [1, 5]. Less than 10% of T13 infants survive the first year of life [1]. Patients with T13 exhibit the unusual phenotype of midline facial and central nervous system defects, such as cleft lip and palate, single orbit, and alobar holoprosencephaly, small ears, and malformation. In addition to this, these patients often have congenital heart disease, abnormalities of the liver, kidneys, lungs, and pancreas [1].

Trisomy patients have needed the support of a wide range of medical professionals, since childhood [10]. The clinical manifestations of these disorders range from relatively innocuous deviations to life-threatening consequences. A trisomy child should have checkup and treatment from several medical professionals, including developmental pediatrician, cardiologist, ophthalmologist, orthopedic specialist, physical and occupational therapist, speech-language therapist, as well as audiologist [11, 12]. These can result in a significant economic burden for both patients and their families.

Currently, prenatal screening via serum screening tests in pregnant woman are included in Thailand's Universal Health Coverage, which covers approximately 80% of the Thai population, and the Thai government intends to expand the prenatal screening test policy to all pregnant women in the near future [13]. Aside from this, more effective methods of detecting fetal cell-free DNA (cfDNA) in maternal plasma, a process known as non-invasive prenatal testing (NIPT), have been developed in recent years, allowing for the detection of embryonic chromosomal abnormalities [14]. Before enforcing prenatal screening for all pregnant women, governments ought to make sure the tests are affordable. However, prenatal screening for T21 patients has only been evaluated financially in a small number of Thai studies. In analysis of serum screening conducted in 2010, Pattanaphesaj et al used a lifetime cost of T21 to be 65,045 USD [15]. From this premise, in 2017, Oraluck et al employed serum screening and NIPT costs across the lifespan of T21 patients from a study by Pattanaphesaj et al in their calculations [16]. In contrast to the above studies, economic evaluation of serum screening and NIPT was undertaken by Wanapirak et al, using lifetime cost data from the United States, which was 583,144 USD [17].

Until recently, no study had been explored the lifetime cost of patients with T13, T18, and T21 in Thailand. Consequently, the objective of this study was to estimate the lifetime cost of patients with T13, T18, and T21 at Siriraj Hospital, Thailand's largest teaching and tertiary hospital, where a wide range of medical professionals and cutting-edge technology are available to care for patients with these conditions. Our study could provide up-to-date and locally-relevant lifetime costs of patients with T13, T18, and T21 which are necessary to be applied in the cost-effectiveness analysis of prenatal screening tests.

## Materials and methods

### Study design

A prevalence-based approach was applied to estimate the economic burden of Thai patients with T13, T18, and T21 using a societal perspective which covered all costs incurred by society

i.e., direct medical, direct non-medical and indirect costs. Direct medical costs included service fee, drug and medical devices for trisomy treatment and co-morbidities, laboratory diagnosis, radiology examination, rehabilitation, operations, and other services such as dental care, psychology, and blood transfusion. The hospital electronic database was retrospectively screened to identify patients with T13, T18, and T21 and calculate direct medical costs for a one-year period in 2016. Direct non-medical costs i.e., transportation and meal expenses incurred by patients and their families during outpatient and inpatient visits were calculated by the average number of outpatient care visits and length of stay (LOS) per case analyzed from the hospital electronic database multiplied with the unit costs of transportation per one round trip and three meals obtained from the standard cost list for health technology assessment (HTA). The list contains the reference unit cost data of medical services and those incurred by patients receiving treatment in Thailand [18] which are commonly used in cost analysis and better reflect costs nowadays in Thailand. Moreover, indirect costs or productivity loss of the patients and caregivers were estimated using a human capital approach calculated by multiplying the number of years lost from work with an annual average of the Gross Domestic Product (GDP) per patient [19].

## Study population

The inpatient and outpatient care data of patients with T13, T18, and T21 for direct medical cost calculation in 2016 were retrospectively retrieved from the Siriraj Informatics and Data Innovation Center (SiData+), Faculty of Medicine, Siriraj Hospital. The International Classification of Disease, the Tenth Revision, Clinical Modification (ICD-10-CM) codes used to identify patients with T13, T18, and T21 were as follows: Q90.0, Trisomy 21 without mosaicism (meiotic nondisjunction); Q90.1, Trisomy 21 with mosaicism (mitotic nondisjunction); Q90.2, Trisomy 21 translocation; Q90.9, Down syndrome not otherwise specified; Q91.0, Trisomy 18 without mosaicism (meiotic nondisjunction); Q91.1, Trisomy 18 with mosaicism (mitotic nondisjunction); Q91.2, Trisomy 18 translocation; Q91.3, Trisomy 18 not otherwise specified; Q91.4, Trisomy 13 without mosaicism (meiotic nondisjunction); Q91.5, Trisomy 13 with mosaicism (mitotic nondisjunction); Q91.6, Trisomy 13 translocation; Q91.7, Trisomy 13 not otherwise specified. Patient data from all wards and departments were obtained except trauma department which might not be related to trisomy patients' co-morbidities. For personal identifiers, assigning pseudonyms were used instead of patient's name and personal data. All raw data were stored in the principal investigator's encrypted computer with password protection. Principal investigator was the only person who could get access and analyze the raw data and other researchers could see only summary tables or charts. The data would be stored in the computer until the study is published. The ethics approval was granted by the Siriraj Institutional Review Board (SIRB) (MU-MOU COA 657/2021), and the requirement for informed consent was waived by the SIRB committee.

## Cost estimation

**Direct medical and non-medical costs.** Direct costs were calculated using the parameters as indicated in Table 1. Total annual direct medical costs of patients with T13, T18, and T21 were classified by outpatient and inpatient care visits as well as age groups. Patients were divided into three groups based on their age i.e., children (0–14 years), adults (≥15 years), and all patients. Even though the effective age of consent in Thailand is 18 years old, we applied 15 years old as the cut-off, since hospitals in Thailand routinely provide care for pediatric patients at age ranging from newborn to 15 years. Moreover, direct non-medical costs included transportation and meal expenses calculated by the average number of outpatient visits and LOS

**Table 1. Parameters used for cost calculations.**

| Type of costs | Parameters | Sources |
|---|---|---|
| Direct medical costs | Service fee | Hospital database |
| | Drug and medical devices | Hospital database |
| | Laboratory diagnosis | Hospital database |
| | Radiology examination | Hospital database |
| | Rehabilitation | Hospital database |
| | Operations | Hospital database |
| | Others (dental care, psychology, blood transfusion) | Hospital database |
| Direct non-medical costs | Unit cost of transportation | [18] |
| | Unit cost of meals | [18] |
| | Average number of outpatient visits | Hospital database |
| | Average length of stay | Hospital database |
| Indirect costs | GDP per capita per year | [23] |
| | Annual income growth rate | [24] |
| | Discount rate | [19] |
| | Survival of patients with T13 | [1] |
| | Survival of patients with T18 | [1] |
| | Survival of patients with T21 | [2] |

multiplied with the unit cost of transportation per one round trip or the unit cost of three meals. We assumed that one caregiver would go to outpatient care and visit inpatient care every day during LOS. The average number of outpatient visits and LOS were estimated from the aforementioned hospital database, while the unit costs of transportation and meals were obtained from the standard cost list for HTA containing the reference unit costs for direct medical, direct non-medical, and indirect cost calculation in Thailand [18]. Consumer Price Index (CPI) was used to adjust all costs from 2016 to 2022. The costs were also converted from Thai baht (THB) to US dollars (USD) using the exchange rate of 38.08 baht per one USD (2022).

**Indirect cost.** According to the Thai HTA guidelines, the human capital approach was applied to calculate indirect costs or productivity loss of both caregivers and adult patients with T21 as well as caregivers for patients with T13 and T18 [19]. Productivity is defined as the annual average per capita of the GDP and Gross National Income (GNI) [19]. To choose between GDP and GNI, it depends upon each country's economic system. It is recommended that GNI should be used if there is a lot of investment and a lot of foreign employees in the country [19]. According to the Thailand Migration Report 2019, migrant workers contributed only 4.3–6.6% of Thailand's GDP [20], thus labor productivity in this study was relied on the average GDP per worker in 2021. In this study, we calculated the productivity loss of both caregivers and T21 patients with the average expected survival of 50 years [2], while we considered the productivity loss of only caregivers for T13 and T18 patients who had average survival of only one year according to published studies [1, 21, 22]. Therefore, productivity loss of caregivers for T21 patients was estimated by the average expected survival of 50 years, since caregivers have to leave their jobs for caring multiplied by the Thai GDP per capita per year (7,233.4 USD) [23], whereas that of patients with T21 was estimated by the average expected working year of T21 patients (35 years) i.e., average survival years (50 years) minus working age adults at 15 years multiplied by the Thai GDP per capita per year. The summation of productivity loss was divided by 35 years for T21 patients and 50 years for their caregivers to generate an annual productivity loss.

To calculate the productivity loss of T21 patients for 35 years and caregivers for 50 years, we calculated expected income each year with the increasing rate of 4%, which was obtained from an annual income growth rate during 2012–2022 in Thailand [24]. Based on the recommendation from the Thai HTA guidelines, since cost values are different in different time periods, future values of total expected income or productivity loss during 35 years (FV) should be adjusted to their present values (PV) using an annual discount rate of 3% based on this formula: $PV = FV \times [1/(1+r)^n$, where PV = present value, FV = future value, r = discount rate, and n = each year in the future [19]. In addition, the lifetime costs were calculated by the summation of direct medical costs, direct non-medical costs and indirect costs of caregivers during one year for T13 and T18 patients, 35 years for T21 patients and 50 years for T21 caregivers.

## Statistical analysis

Statistical analysis was performed using the Microsoft Excel 2019 (Microsoft, WA, USA). Demographic data were analyzed using descriptive statistics. Costs were presented as mean with standard deviation (SD) to represent the cost burden and as median with minimum and maximum as well as interquartile range (IQR) to represent the probable anticipated cost for an individual. Besides, generalized linear models (GLMs) with gamma distribution were applied to investigate the relationship between annual total direct medical cost and confounding factors i.e., age, female gender, number of outpatient or inpatient visits, and having CHD [22]. Statistical differences between T21 with CHD and without CHD were also calculated using the nonparametric Mann-Whitney U-test for non-normally distributed data, and the parametric equivalent Student's t test for normally distributed data. A p-value less than 0.05 was regarded as being statistically significant.

## Results

### Demographic characteristics of patients

Table 2 presents the demographic characteristics of T13, T18, and T21 patients. Total of 377 patients with T13 (5 patients), T18 (7 patients), and T21 (365 patients) were included in our analysis. All patients with T13 (5 patients) and T18 (7 patients) received both outpatient and inpatient care services, whereas 241 patients with T21 had outpatient visits and 124 received inpatient care. The age of those affected with T21 ranged from 0 to 59 years, with a mean age of 10.4 ± 9.6 years and a median age of 7 years. The age of T13 patients ranged from 0 to 14 years, with a mean age of 4.8 ± 5.4 years and a median age of 3 years, while T18 patients ranged in age from 0 to 14 years, with a mean age of 4.4 ± 6.6 years and a median age of 1 years. Approximately 55% of T21 patients were female and 45% were male. T13 and T18 patients were 80% and 100% female, respectively. Approximately 86% of T18 patients and 60% of T21 patients reported ever having had a cardiac problem. Average outpatient visits for T13, T18, and T21 patients were 9±9.6, 6±7.5, 14.2±13 per year, respectively. Average duration of inpatient stay 55 ± 109 days for T13 patients, 46 ± 54 for T18 patients, and 88 ± 211 days for T21 patients.

### Total direct medical cost

Mean annual direct medical costs for patients with T13, T18, and T21 receiving outpatient care ranged from 183 to 655 USD per patient. For inpatient care, mean annual direct medical costs ranged from 2,507–14,791 USD per patient. Mean and median direct medical costs increased with age for patients with T21 (Table 3).

Fig 1 shows the percentage of total direct medical costs classified by type of costs for outpatient and inpatient care. For outpatient care, drug and medical device costs were large

**Table 2. Demographic characteristics of trisomy patients.**

| Demographic characteristics | Outpatient care (N = 253) | | | Inpatient care (N = 136) | | |
|---|---|---|---|---|---|---|
| | Trisomy 13 (N = 5) | Trisomy 18 (N = 7) | Trisomy 21 (N = 241) | Trisomy 13 (N = 5) | Trisomy 18 (N = 7) | Trisomy 21 (N = 124) |
| Age, years | | | | | | |
| Mean ± SD | 4.8±5.4 | 4.4±6.6 | 10.4±9.6 | 4.8±5.4 | 4.4±6.6 | 9.9±10.5 |
| Median (IQR) | 3 (3–4) | 1 (0–8) | 7 (4–13) | 3 (3–4) | 1 (0–8) | 7 (4–12) |
| Range | 0–14 | 0–14 | 0–59 | 0–14 | 0–14 | 0–59 |
| Children <14 years | 5 (100) | 7 (100) | 191 (79.3) | 5 (100) | 7 (100) | 106 (85.5) |
| Adult > = 15 years | 0 (0) | 0 (0) | 50 (20.7) | 0 (0) | 0 (0) | 18 (14.5) |
| Sex | | | | | | |
| Female | 4 (80) | 7 (100) | 132 (54.8) | 4 (80) | 7 (100) | 66(53.2) |
| Male | 1(20) | 0 (0) | 109 (45.2) | 1(20) | 0 (0) | 58(46.8) |
| Congenital heart disease | | | | | | |
| Yes | 0 (0) | 6 (85.7) | 127 (52.7) | 0 (0) | 6 (85.7) | 74 (59.7) |
| No | 5 (100) | 1 (14.3) | 114 (47.3) | 5 (100) | 1 (14.3) | 50 (40.3) |
| Number of Visits, N | | | | | | |
| Mean ± SD | 9±9.6 | 6±7.5 | 14.2±13 | 2.4 ± 2.2 | 2.2 ± 1.5 | 1.5 ± 1.4 |
| Median (IQR) | 6 (3–10) | 3(1.5–6.5) | 10 (6–18) | 1 (1–3) | 2 (1.25–2) | 1 (1–2) |
| Range | 1–25 | 1–22 | 1–110 | 1–6 | 1–5 | 1–10 |
| Length of Stay, Days | | | | | | |
| Mean ± SD | NA | NA | NA | 55 ± 109 | 46 ± 54 | 88 ± 211 |
| Median (IQR) | NA | NA | NA | 6 (1–18) | 17 (9–90) | 23 (15–66) |
| Range | NA | NA | NA | 1–248 | 4–116 | 3–1,900 |

SD; Standard Deviation, IQR; Interquartile Range, NA; Not Applicable

contributors of the total direct medical costs for T13 and T21 patients, while laboratory costs were considerable for T18 patients (Table 4). For inpatient care, drug and medical device costs were the greatest for T13, whereas drug and medical device cost were the highest for T13, while service fee and operation costs were the highest for T18 and T21, respectively (Table 5).

## Factors associated with total direct medical cost

Table 6 demonstrates the factors associated with total direct medical costs using the GLM regression analysis. For patients with T13, females had significantly lower direct medical costs compared to males. Higher number of outpatient visits and longer LOS could significantly

**Table 3. Total annual direct medical costs of patients with T13, T18, and T21 (USD per patient).**

| Annual direct medical cost | Outpatient care | | | | | Inpatient care | | | | |
|---|---|---|---|---|---|---|---|---|---|---|
| | Trisomy 13 | Trisomy 18 | Trisomy 21 | | | Trisomy 13 | Trisomy 18 | Trisomy 21 | | |
| | Children (N = 5) | Children (N = 7) | Children (N = 191) | Adults (N = 50) | All patients (N = 241) | Children (N = 5) | Children (N = 7) | Children (N = 106) | Adult (N = 18) | All patients (N = 124) |
| Mean | 183 | 321 | 364 | 1,768 | 655 | 14,791 | 3,815 | 2,376 | 3,278 | 2,507 |
| SD | 176 | 379 | 450 | 8,464 | 3,887 | 23,821 | 5,468 | 4,540 | 5,037 | 4,605 |
| Median | 121 | 221 | 221 | 231 | 228 | 780 | 1020 | 851 | 1317 | 919 |
| Min | 10 | 36 | 2 | 11 | 2 | 290 | 214 | 70 | 54 | 54 |
| Max | 432 | 1134 | 3,519 | 60,058 | 60,058 | 55,559 | 14,244 | 27,036 | 18,427 | 27,036 |
| Interquartile Range (IQR) | 59–294 | 92–336 | 136–424 | 91–604 | 130–467 | 727–16,596 | 824–4,482 | 530–2,165 | 563–3,404 | 523–2,201 |

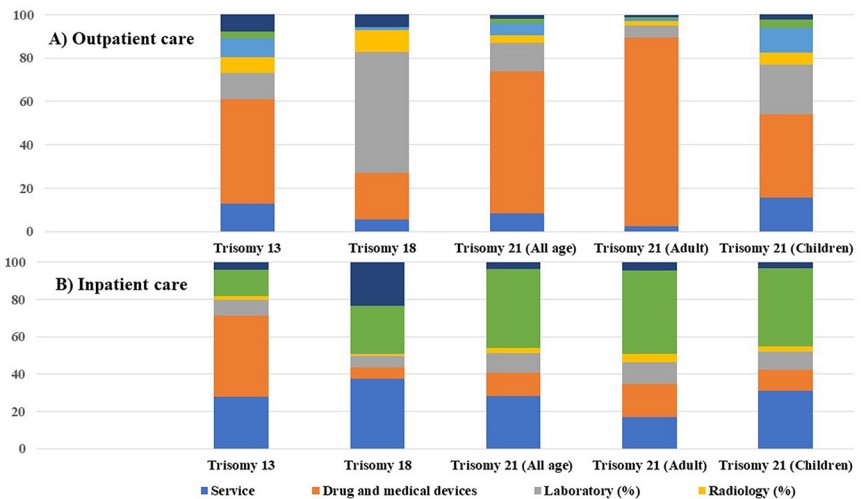

**Fig 1. Percentage of total direct medical costs for patients with T13, T18, and T21.**

increase direct medical costs in all trisomy patients. Patients with older age significantly consumed higher direct medical costs in outpatient care, but not inpatient care for patients with T18 and T21. Moreover, T21 patients without CHD significantly had lower direct medical

**Table 4. Total annual direct medical costs for patients with T13, T18, T21 classified by type of costs (USD) for outpatient care.**

| Type of costs | Outpatient care | | | | | | | | | | | | | | |
|---|---|---|---|---|---|---|---|---|---|---|---|---|---|---|---|
| | Trisomy 13 | | | Trisomy 18 | | | Trisomy 21 | | | | | | | | |
| | Children | | | Children (N = 7) | | | Children (N = 191) | | | Adults (N = 50) | | | All ages (N = 241) | | |
| | Mean (SD) | Median (Range) | IQR | Mean (SD) SD | Median (Range) | IQR | Mean (SD) | Median (Range) | IQR | Mean (SD) | Median (Range) | IQR | Mean (SD) | Median | Median (Range) |
| **Service** | 23.8 (20.5) | 16.4 (7.1–59.4) | 14.8–21.4 | 17.6 (17.5) | 14.2 (2.4–47.5) | 2.5–26.9 | 57 (68) | 38 (2.4–457) | 17.1–67 | 44.8 (43.5) | 28 (2.4–175.2) | 16.7–46.5 | 54.5 (63.8) | 34.2 (2.4–457) | 17–66.4) |
| **Drug and medical devices** | 146.6 (203.2) | 55.6 (4.8–379.4) | 30.2–217.5 | 69.3 (72.6) | 53.9 (5.7–206.6) | 11–98.5 | 150 (346) | 35.9 (0.3–2,725.4) | 13–129.9 | 1708 (8895) | 76.9 (8.3–59770.2) | 19.5–210.3 | 464.4 (4021) | 41 (0.3–59,770.2) | 14.2–155.5 |
| **Laboratory** | 27.8 (22.9) | 30.6 (2.6–47.5) | 11.3–47.1 | 178.3 (190.8) | 147.4 (17.8–578.7) | 56.7–195.3 | 86.6 (78.1) | 68.6 (2.7–461.3) | 36.7–111.5 | 105.3 (150.8) | 63.6 (2.1–968.1) | 35–116.8 | 90.4 (97.3) | 68.1 (2.1–968.1) | 35.4–112.1 |
| **Radiology** | 32.3 (36.4) | 32.3 (6.6–58.1) | 19.5–45.2 | 46.3 (67.5) | 23 (6.6–166.2) | 9.8–26.1 | 30.4 (59) | 13.3 (6.6–527.1) | 6.6–26.5 | 61.9 (181.9) | 26.5 (6.6–962.4) | 6.6–34.8 | 35.8 (92.6) | 15.4 (6.6–962.4) | 6.6–29.2 |
| **Rehabilitation** | 40.9 (24.3) | 40.9 (23.7–58.1) | 32.3–49.5 | 23.7 (NA) | 23.7 | NA | 80.8 (117) | 42.5 (5.3–871.9) | 21.2–102.2 | 95.5 (119) | 35.8 (2.7–345) | 23.9–152.6 | 82.2 (117.3) | 42.5 (2.7–871.9) | 21.2–104.8 |
| **Operation** | 26.4 (NA) | 26.4 | NA | 0 | 0 | 0 | 38.7 (87.1) | 15.9 (8–694.2) | 8–31.8 | 43.5 (67) | 15.9 (2.7–233.3) | 8–33.2 | 39.5 (83.6) | 15.9 (2.7–694.2) | 8–31.8 |
| **Others** | 73.1 (NA) | 73.1 | NA | 44.9 (57.9) | 21.1 (2.6–110.8) | 11.9–66 | 21.1 (24.7) | 11.9 (3.7–173.8) | 5.3–26.5 | 89.1 (202.2) | 11.9 (3.7–704.4) | 6.5–604 | 30.5 (79.5) | 11.9 (3.7–704.4) | 5.3–28.7 |

SD; Standard Deviation, IQR; Interquartile Range

Table 5. Total annual direct medical costs for patients with T13, T18, T21 classified by type of costs (USD) for inpatient care.

| Type of costs | Inpatient care | | | | | | | | | | | | | | |
|---|---|---|---|---|---|---|---|---|---|---|---|---|---|---|---|
| | Trisomy 13 | | | Trisomy 18 | | | Trisomy 21 | | | | | | | | |
| | Children | | | Children (N = 7) | | | Children (N = 191) | | | Adults (N = 50) | | | All ages (N = 241) | | |
| | Mean (SD) | Median (Range) | IQR | Mean (SD) | Median (Range) | IQR | Mean (SD) | Median (Range) | IQR | Mean (SD) | Median (Range) | IQR | Mean (SD) | Median | Median (Range) |
| **Service** | 4,135.7 (6,716.4) | 261.7 (63.2–15,649.7) | 126.1–4,577.9 | 1,430.8 (1,642.1) | 861.4 (38.3–4,363.7) | 321.3–1,954.2 | 739.9 (2096.8) | 244.7 (44.6–16,248.2) | 93.1–555.4 | 555.2 (956.8) | 164.6 (39.8–3,302) | 63.7–493.7 | 713.1 (1970.8) | 232.8 (39.8–16,248) | 90–548.1 |
| **Drug and medical devices** | 6,399.2 (11,976.7) | 58.2 (13.7–27,560.9) | 14.5–4,348.9 | 225.6 (324.4) | 59.1 (7.2–808.5) | 15.2–329.3 | 267 (968.3) | 30.1 (0.2–7,936.2) | 13.4–138 | 615.4 (1444.5) | 55.5 (2.3–4,583.9) | 14.3–185.1 | 315.2 (1046.5) | 30.5 (0.2–7,936.2) | 13.5–141.5 |
| **Laboratory** | 1573.6 (1880.2) | 1050 (13.3–4,181.3) | 309.9–2,313.7 | 287.6 (373.4) | 140 (2.1–897) | 15.6–383.3 | 260.2 (712.5) | 59.6 (1.6–5,365.5) | 20.5–199 | 440.1 (913.3) | 59.3 (2.1–3,019.8) | 13.9–258.8 | 285.9 (742.4) | 59.3 (1.6–5,366.5) | 19.1–220.7 |
| **Radiology** | 544.8 (748.9) | 216 (16.5–1,402) | 116.3–808.9 | 67.3 (68.8) | 63 (6.6–136.7) | 9–121.3 | 127.7 (279) | 39.3 (4.2–,1795) | 13.3–84.9 | 402.4 (476.5) | 191.2 (9.8–1,038) | 34.4–799.5 | 156.6 (311.5) | 39.3 (4.2–1,795) | 13.3–114.4 |
| **Rehabilitation** | 0 (0) | 0 (0) | 0 (0) | 101.3 (NA) | 101.3 | NA | 81.6 (68.8) | 94 (11.9–182.1) | 22–98.2 | 0 (0) | 0 (0) | 0 (0) | 81.6 (68.8) | 94 (11.9–182.1) | 22–98.2 |
| **Operation** | 2071.1 (2749.6) | 586.7 (35–6,477) | 198.7–3,058.1 | 1162.6 (2322.1) | 37.3 (6.6–5,302.2) | 6.8–459.8 | 1208.9 (1559.9) | 503.6 (5.3–9,486.7) | 345.5–1,588.8 | 1648.2 (1996) | 918.1 (10.6–6,869.4) | 416.1–1766.4 | 1277.1 (1631.9) | 509.7 (5.3–9,486.7) | 357.9–1,588.8 |
| **Others** | 997.9 (1484.8) | 288.5 (0.8–2,704.3) | (144.7–1,496.4) | 888.8 (1943.2) | 77.5 (3.2–4847.8) | 13–270.9 | 87.7 (236.1) | 5.3 (0.8–1,832.9) | 1.6–68.8 | 762 (301.6) | 5 (0.8–2,320) | 0.8–116.5 | 106.7 (317.7) | 5 (0.8–2,320) | 1.6–72.2 |

SD; Standard Deviation, IQR; Interquartile Range

**Table 6. Factors associated with total direct medical costs.**

| Trisomy patients | Outpatient Care | | | | Inpatient Care | | | |
|---|---|---|---|---|---|---|---|---|
| | Parameters | Value | Standard Error | P-value | Parameters | Value | Standard Error | P-value |
| T13 | Intercept | 8.29 | 0.33 | <0.01 | Intercept | 10.18 | 0.19 | <0.01 |
| | Sex (Female) | -3.36 | 0.54 | <0.01 | Sex (Female) | -1.48 | 0.27 | <0.01 |
| | Age | 0.27 | 0.04 | <0.01 | Age | 0.31 | 0.02 | <0.01 |
| | Number of visit | 0.15 | 0.02 | <0.01 | Length of stay | 0.02 | 0.00 | <0.01 |
| T18 | Intercept | 7.66 | 0.43 | <0.01 | Intercept | 11.30 | 0.59 | <0.01 |
| | No CHD | -3.64 | 2.12 | 0.09 | No CHD | 1.81 | 1.54 | 0.24 |
| | Age | 0.03 | 0.04 | 0.48 | Age | -0.05 | 0.10 | 0.59 |
| | Number of visit | 0.28 | 0.10 | <0.01 | Length of stay | 0.01 | 0.00 | 0.05 |
| T21 | Intercept | 8.28 | 0.17 | <0.01 | Intercept | 10.50 | 0.14 | <0.01 |
| | Sex (Female) | -0.01 | 0.13 | .931 | Sex (Female) | 0.41 | 0.14 | <0.01 |
| | No CHD | -0.37 | 0.13 | <0.01 | No CHD | -0.52 | 0.15 | <0.01 |
| | Age | 0.07 | 0.01 | <0.01 | Age | 0.01 | 0.01 | 0.33 |
| | Number of visit | 0.05 | 0.01 | <0.01 | Length of stay | 0.00 | 0.00 | <0.01 |

CHD; congenital heart disease

costs than those with CHD. Similarly, for outpatient care, only T21 adult patients with CHD had significantly higher mean annual direct medical costs than those without CHD. In addition, all T21 patients and pediatric patients with CHD receiving inpatient care incurred significantly higher total direct medical costs (Table 7).

**Table 7. Total annual direct medical costs incurred with an 'ever' diagnosis of congenital heart disease (USD) for trisomy 21 patients.**

| Age group | CHD | | | | | No CHD | | | | | | P-value |
|---|---|---|---|---|---|---|---|---|---|---|---|---|
| | Mean | SD | Median | Range | IQR | | Mean | SD | Median | Range | IQR | |
| **Outpatient care** | | | | | | **Outpatient care** | | | | | | |
| All ages (N = 127) | 923 | 5,324 | 241 | 8–60,058 | 134–573 | All ages (N = 114) | 357 | 570 | 216 | 2–4,962 | 122–374 | 0.26 |
| Adult (N = 6) | 4,558 | 14,824 | 639 | 48–60,058 | 220–1,496 | Adult (N = 34) | 455 | 964 | 167 | 11–4,962 | 84–347 | <0.01 |
| Children (N = 111) | 399 | 544 | 221 | 8–3,519 | 123–480 | Children (N = 180) | 315 | 267 | 238 | 2–1,514 | 133–385 | 0.78 |
| **Inpatient care** | | | | | | **Inpatient care** | | | | | | |
| All age (N = 74) | 3,378 | 5,553 | 1,393 | 54–27,036 | 659–3,444 | All age (N = 50) | 1219 | 2111 | 762 | 70–13,627 | 467–1,171 | <0.01 |
| Adult (N = 8) | 3,475 | 6,172 | 1,360 | 54–18,427 | 572–2,184 | Adult (N = 10) | 3120 | 4271 | 1229 | 434–13,627 | 587–1,404 | 0.89 |
| Children (N = 66) | 3,366 | 5,525 | 1,393 | 408–27,036 | 659–3,444 | Children (N = 40) | 744 | 481 | 618 | 70–1,960 | 445–138 | <0.01 |

CHD; congenital heart disease, SD; Standard Deviation, IQR; Interquartile Range

**Table 8. Lifetime costs of patients with T13, 18 and 21.**

| Type of cost (USD) | Trisomy 13 | Trisomy 18 | Trisomy 21 |
|---|---|---|---|
| **Annual direct medical cost (Mean ± SD)** | | | |
| Outpatient care | 183±176 | 321±379 | 655± 3,887 |
| Inpatient care | 14,791± 23,821 | 3,815±5,468 | 2,507±4,605 |
| Total annual direct medical cost | 14,974±23,997 | 4,136±5,847 | 3,162± 8,492 |
| **Annual direct non-medical cost (Mean ± SD)** | | | |
| Transportation | 561±1,040.0 | 456±539 | 903±1,964 |
| Food | 127±213 | 99±118 | 206±387 |
| Total annual direct non-medical cost | 688±1,253 | 555±658 | 1,109±2,351 |
| **Annual indirect cost (USD)** | | | |
| Productivity loss of trisomy patients | - | - | 10,137 |
| Productivity loss of caregivers | 7,233 | 7,233 | 9,255 |
| Total annual indirect cost | 7,233 | 7,233 | 19,392 |
| **Annual total cost of illness (USD)** | 22,715 | 11,924 | 23,663 |
| **Survival (years)** | 1 | 1 | • 50 years for caregivers<br>• 35 years for T21 patients |
| **Lifetime cost (USD)** | 22,715 | 11,924 | 1,031,111 |

## Lifetime costs for trisomy patients

Lifetime costs for patients with T13, 18 and 21 are shown in Table 8. Of all trisomy patients, total annual average direct medical costs of patients with T13 was the highest, while those with T21 had the highest total annual average direct non-medical and indirect costs. Consequently, patients with T21 had the highest total annual cost of illness. However, when considering the costs incurred in a lifetime period, T21 patients consumed the highest (1,031,111 USD), followed by T13 (22,715 USD) and T18 (11,924 USD) patients.

## Discussion

This study was the first to investigate lifetime costs incurred by T13, T18, and T21 patients in a Thai tertiary hospital. We explored direct medical, direct non-medical, and indirect costs incurred by trisomy patients with all ages in a single study. Typically, either children or adults have been the subjects of cost studies. Our study showed the annual average direct medical costs for pediatric patients with T13 and T18 were 14,974 and 4,136 USD, respectively, which were lower than a previous study from the US revealing mean annual hospital charges of 30,021 USD and 39,537 USD [25]. Similarly, mean annual direct medical costs (2,740 USD) in this study was much lower compared to the studies in Australia [26], the US [11, 25, 27]. and Korean [28]. This can be explained that direct healthcare costs incurred in Thai hospitals are much lower compared to those in developed countries.

In addition, T21 adults aged older than 15 years in our study (5,046 USD) consumed annual average direct medical cost less than that in the US (18,241 USD) based on the Nationwide Inpatient Sample database [29]. The annual mean direct medical cost was much higher among Down syndrome patients with dementia aged 45–89 (35,011 USD) compared to those without dementia (24,401 USD), due to higher inpatient services care, multimorbidity, and more primary care physician and emergency department visits [25]. However, our study showed similar average direct medical costs among T21 adults to the study conducted in a hospital in Taiwan (5,006 USD) [30].

Besides, our study revealed that T21 patients with CHD in adult groups receiving outpatient care and pediatric patients receiving inpatient care had significantly higher mean annual direct

medical costs than those without CHD in all patient groups. This is consistent with Boulet SL et al study in the Western Australia suggesting that the mean and median costs for Down syndrome newborns with CHD were five to seven times more than for Down syndrome infants without CHD [26]. Similarly, mean annual inpatient cost ranged between 9,706 USD and 109,059 USD for Down syndrome patients with CHD in the US [11].

Furthermore, the average annual direct healthcare costs among Down Syndrome patients aged 0 to 25-year-old in the Western Australia was 4,287 USD, and those costs dropped with age, as hospital, physician, and treatment consumption reduced [10]. In contrast, our study demonstrated that the mean and median costs for T21 patients increased with age. Moreover, the results of GLM regression analysis indicated that trisomy patients with older age significantly incurred higher direct medical costs in outpatient care. It was noticed that Down syndrome patients without CHD in all age groups had similar outpatient costs, but adult groups spent more inpatient costs compared to the pediatric group. This situation might be due to a difference in costs of health care for T21 pediatric and adult patients. In previous study from Geelhoed, speech therapy and occupational therapy were the most prevalent treatments for T21 pediatric patients, with mean annual costs of 1,878 USD and 1,505 USD, respectively [10], while the patients were not charged for these treatments in our study. As a result, the greater use of drugs and medical devices by adults compared to pediatric patients contributed to the rise in adult healthcare costs.

Rendering care for trisomy patient results in increasing direct non-medical and indirect costs for caregivers. Our study estimated direct non-medical cost for trisomy care ranging from 500 to 1,100 USD per year and indirect cost from productivity loss of trisomy patients and caregivers ranging from 7,000 to 19,000 USD per year. These costs were lower compared to the study from Stabile and Allin indicating that the family costs for a child with a disability or chronic condition ranged between 20,000 and 60,000 USD per year [31]. Furthermore, our study found that caregivers were more likely to be female. This is in line with the study of Martınez-Valverde et al in Mexico suggesting that mothers of T21 pediatric patients had to leave their jobs for caring them and this might cause a negative effect on family income [12].

Notably, our study revealed the lifetime cost for trisomy patients ranging from 20,000 to 1,100,000 USD depending on the average survival year of the patients, which seemed to be lower compared to the economic costs of childhood disability ranging from 41,000 to 4,300,000 USD according to a literature review [32]. Nevertheless, compared to lifetime economic burden of T21 patients in Asia countries like China (55,000 USD) [33] and Thailand (72,000 USD) [15], our lifetime economic burden of T21 patients was higher for the reason that our direct medical costs data were collected from the largest teaching and tertiary hospital where leading-edge and high cost technology are available to care for these patients and we assumed that T21 patients would receive treatment and have productivity loss for 35 years. Thus, this may overestimate the lifetime cost of T21 patients in our study.

To the best of our knowledge, this is the first study to demonstrate both direct medical, direct non-medical and indirect costs of patients with T13, T18, and T21 in Thailand which could fill in the gap with the latest lifetime costs of these patients specifically in local Thai context. This information would be necessary to be applied in the cost-effectiveness analysis of prenatal screening tests which can be used for policy decision making whether the tests should be included in the benefit package of the Universal Health Coverage. In addition, the results from this study can help support the Thai government to implement the prenatal screening test policy to all pregnant women in the near future. However, this research was limited by the fact that the data were obtained from a single hospital and direct non-medical costs were calculated using standard costing methods. This may result in the underestimation of the cost for trisomy care in Thailand. However, the treatments of trisomy patients such as specific

operation for CHD, gastrointestinal abnormalities from specialist and intensive care unit are only available at university hospital such as Siriraj hospital. Therefore, the results from this study cannot be generalized to Thailand as a whole. Notwithstanding these limitations, this research provided not only the evidence of the economic burden of T13, T18, and T21, but also the proper economic assessment for these diseases in local Thai context. Future studies should be further investigated on collecting more direct medical cost data from other hospitals and direct non-medical cost data from interviewing trisomy patients and their caregivers.

## Conclusion

Our study demonstrates direct medical, direct non-medical, indirect costs as well as the lifetime costs of Thai patients with T13, T18, and T21 in a tertiary hospital based on a societal perspective. The results confirm that there is a significant economic burden of patients with T13, T18, and T21 in Thailand. The findings from this study could provide up-to-date direct and indirect cost information for trisomy patients especially in Thai context which would be the best available evidence for conducting economic evaluation of prenatal screening for policy makers which can help promote the prenatal screening policy in the future.

## Acknowledgments

The authors gratefully acknowledge Assoc. Prof. Panutsaya Tientadakul, Head of the Department of Clinical Pathology, Faculty of Medicine Siriraj Hospital, Mahidol University for her support.

## Author Contributions

**Conceptualization:** Preechaya Wongkrajang, Usa Chaikledkaew.

**Formal analysis:** Preechaya Wongkrajang.

**Investigation:** Preechaya Wongkrajang.

**Methodology:** Preechaya Wongkrajang, Jiraphun Jittikoon, Wanvisa Udomsinprasert, Pattarawalai Talungchit, Usa Chaikledkaew.

**Resources:** Preechaya Wongkrajang.

**Supervision:** Usa Chaikledkaew.

**Writing – original draft:** Preechaya Wongkrajang, Jiraphun Jittikoon, Wanvisa Udomsinprasert, Pattarawalai Talungchit, Usa Chaikledkaew.

**Writing – review & editing:** Preechaya Wongkrajang, Jiraphun Jittikoon, Wanvisa Udomsinprasert, Pattarawalai Talungchit, Usa Chaikledkaew.

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
