## [Decision Letter · Decision Letter 0]

27 Apr 2023

PONE-D-23-05563Economic Cost of Patients with Trisomy 13, 18, 21 in ThailandPLOS ONE

Dear Dr. Chaikledkaew,

Thank you for submitting your manuscript to PLOS ONE. After careful consideration, we feel that it has merit but does not fully meet PLOS ONE’s publication criteria as it currently stands. Therefore, we invite you to submit a revised version of the manuscript that addresses the points raised during the review process.

This is a cost of disease study and is of interest to general readers. Please kindly address the comments of the 2 reviewers. 

We look forward to receiving your revised manuscript.

Kind regards,

Yee Gary Ang, MBBS MPH

Academic Editor

PLOS ONE

“No”

Reviewers' comments:

Reviewer's Responses to Questions

**Comments to the Author**

1. Is the manuscript technically sound, and do the data support the conclusions?

Reviewer #1: Yes

Reviewer #2: Yes

2. Has the statistical analysis been performed appropriately and rigorously? 

Reviewer #1: No

Reviewer #2: Yes

3. Have the authors made all data underlying the findings in their manuscript fully available?

Reviewer #1: Yes

Reviewer #2: Yes

4. Is the manuscript presented in an intelligible fashion and written in standard English?

Reviewer #1: Yes

Reviewer #2: Yes

5. Review Comments to the Author

Reviewer #1: This study assessed the healthcare use and associated costs for patients with trisomy 13, 18, 21 in Thailand. Overall, the findings of the study benefits to further economic research in Thailand. However, some concerns need to be addressed.

1. The title of the study specified “Economic cost of patients with trisomy 13, 18, 21 in Thailand”. However, only one site with very small sample size (trisomy 13, 18) might not represent the cost for the whole country?

2. The study uses descriptive statistics to provide % cost contribution, costs of outpatient and inpatient. Those costs did not control for other confounders such as age, other co-morbidities, gender, different length of stay. A more robust analysis like GLM regression could provide a more robust result?

3. The data were retrieved via ICD-10-CM. How many years are there for the database?

4. Direct non-medical costs were calculated from standard costing. Why did authors use the standard costing? Need to explain the strength of the standard costing. Does it still reflect direct non-medical costs nowadays?

5. Indirect cost was determined using a human capital approach. The majority of population in this study is children. Is this approach appropriate for those who are not working? In addition, the detail of calculations should be added.

6. Where did 4% (page 6) come from and how did author apply in the formular given?

7. Table 2: what is the unit of annual direct medical cost for OPD and IPD? Is it per visit, per admission, or per patient? Table 1 shows number of visits, and LOS. Authors should classify them by CHD.

8. Drug and medical device are the main component of costs. Do they include drug costs for co-morbidities?

9. Table 6 displays the annual average cost. Indirect costs of each trisomy were equal. Why? Furthermore, indirect cost of patient was equal to that of care giver? Patients were children, but care givers were generally their parents.

10. According to Table 1, age range of patient with trisomy 13 and trisomy 18 was 0-14 years. Table 6. Why is lifetime cost equal to annual total cost for Trisomy 13 and Trisomy 18? Patients can live longer than a year. In addition, how did author calculate lifetime cost? The detail should be added.

11. Authors might display some results using pictures instead of tables.

Reviewer #2: I thank the editor for the opportunity to review this manuscript. I also commend the authors for undertaking this important work to delineate the cost burden faced by trisomy patients and their caregivers. Overall, the paper is sound and clear with the necessary analyses done. However, I do have concerns with the way the results have been presented in the tables and the discussion section. The discussion section needs to be more concise and clearly link the findings back to the existing evidence. While the authors have attempted this, it seems as though the authors are merely listing other studies rather than to make meaningful linkages back to the findings and how it adds value to the topic/field. I have provided some brief comments below and have added more detailed feedback and edits to the attached document.

It seems a little confusing that the first sentence says "cross-sectional study" but the second sentence says it is a "retrospective" study. I would think the authors are saying that the electronic database was retrospectively screened to identify patients with T13, T18, T21 and calculate the direct medical costs for an one-year period. If this is right, please amend as suggested.

Can the authors elaborate briefly how the costs were calculated please? Did the family of the patients provide any cost-related information to this study?

Please elaborate exactly the data fields that were collected to aid in the cost calculation. Maybe including a table of variables that were extracted from the electronic database would be good (unless it is sensitive) to include here so that readers could know what parameters were used to calculate the direct medical costs.

I'm aware 15 is the legal age for consent in Thailand. However, may I clarify that children are treated as adults at 15 years old? May I also just clarify the rationale for setting 15 years old as the cut-off and not 18 years, which is the effective age of consent please?

How was patient data handled? Were they all anonymized data? Suggest to include some information about how the extracted data was handled - i.e. were there any personal identifiers, how was the data stored, how long was it stored, was it password-protected, who had access etc. under the methods section. This is important in ensuring responsible use of patient data in studies.

For all tables, suggest to break them into 2 separate sub-tables for outpatient and inpatient care so that the tables do not look too packed.

Please standardize the number of decimal places/significant figures for p-values. For instance, you can leave it as 2 sig figures. For p-values like this, you can indicate as p <0.01.

Please refer to the attached file for the detailed comments and edits.

Once again, I thank the authors for working on this important topic and I wish them all the best in their work.

6. PLOS authors have the option to publish the peer review history of their article (what does this mean?). If published, this will include your full peer review and any attached files.

Reviewer #1: No

Reviewer #2: No

---

## [Author Response · Author response to Decision Letter 0]

10 Jun 2023

Response to Journal requirements & Reviewer:

Response: Thank you very much. We have already checked that our manuscript meets PLOS ONE’s style requirements. 

Response: According to your suggestion, we have added the ethics statement under “Study population” sub-heading as follows.

 “The ethics approval was granted by the Siriraj Institutional Review Board (SIRB) (MU-MOU COA 657/2021), and the requirement for informed consent was waived by the SIRB committee.” (Study population, line 128-130, page 7)

“No”

Response: We would like to provide financial disclosure as follows: “This study receives funding support from the Health Systems Research Institute (HSRI). The funders had no role in study design, data collection and analysis, decision to publish, or preparation of the manuscript." We also include our amended statements within our cover letter. 

Response: Data cannot be shared publicly because our study obtained data from patients with trisomy 13, 18, and 21 which contained sensitive patient information and there are ethical restrictions on publicly sharing a sensitive data set. Data are available from the Human Research Protection Unit, Faculty of Medicine Siriraj Hospital, Mahidol University (contact via Room 210, 2nd floor, His Majesty the King's 80th Birthday Anniversary 5th December 2007 Building, 2 Wang Lang Road Bangkoknoi, Bangkok 10700 or siethics@mahidol.ac.th) for researchers who meet the criteria for access to confidential data.

Reviewer #1: 

This study assessed the healthcare use and associated costs for patients with trisomy 13, 18, 21 in Thailand. Overall, the findings of the study benefits to further economic research in Thailand. However, some concerns need to be addressed.

Response: Thank you very much for taking your time to review our manuscript. We feel that our manuscript has been much further improved according to your great suggestions. 

1. The title of the study specified “Economic cost of patients with trisomy 13, 18, 21 in 

Thailand”. However, only one site with very small sample size (trisomy 13, 18) might not represent the cost for the whole country?

Response: Thank you very much for your great suggestion. We agreed with your suggestion and we have changed the title as “Economic cost of patients with trisomy 13, 18, 21 in a tertiary hospital in Thailand” (Title, line 1, page 1)

2. The study uses descriptive statistics to provide % cost contribution, costs of outpatient 

and inpatient. Those costs did not control for other confounders such as age, other co-morbidities, gender, different length of stay. A more robust analysis like GLM regression could provide a more robust result?

Response: Thank you very much for very helpful suggestion. We have performed GLM regression analysis and added more sentences in materials and methods, results, and discussion parts. 

“Besides, generalized linear models (GLMs) with gamma distribution were applied to investigate the relationship between annual total direct medical cost and confounding factors i.e., age, female gender, number of outpatient or inpatient visits, and having CHD [22].” (Statistical analysis, line 184-186, page 9)

Table 6 (Results, line 222-232, page 13)

“Moreover, the results of GLM regression analysis indicated that trisomy patients with older age significantly consumed higher direct medical costs in outpatient care.” (Discussion, line 289-291, page 20)

3. The data were retrieved via ICD-10-CM. How many years are there for the database?

Response: Thank you very much for your kind question. One-year data in 2016 were retrieved via ICD-10-CM. We have revised the sentence as follows. 

“A hospital electronic database was retrospectively screened to identify patients with T13, T18, and T21 and calculate direct medical cost associated with treatment for a one-year period in 2016.” (line 98-100, page 5)

4. Direct non-medical costs were calculated from standard costing. Why did authors use 

the standard costing? Need to explain the strength of the standard costing. Does it still reflect direct non-medical costs nowadays?

Response: Thank you very much for raising this point. We have added the below sentences as suggested.

“The list contains the reference unit cost data of medical services and those incurred by patients receiving treatment in Thailand [18] which are commonly used in cost analysis and better reflect costs nowadays in Thailand.” (Study design, line 105-107, page 5-6).

5. Indirect cost was determined using a human capital approach. The majority of population in this study is children. Is this approach appropriate for those who are not working? In addition, the detail of calculations should be added.

Response: Thank you very much for pointing this out. Indirect costs were calculated using a human capital approach which was applied only adult patients with T21 and caregivers for patients with T13 and T18, as this approach is appropriate for those who are working. We have added the sentence below for clarification.

“According to the Thai HTA guidelines, the human capital approach was applied to calculate indirect costs or productivity loss of both caregivers and adult patients with T21 as well as caregivers for patients with T13 and T18 [19].” (Indirect cost, line 152-154, page 8)

More details of calculations have been added under “Indirect cost” sub-heading. (line 151-166, page 8). 

6. Where did 4% (page 6) come from and how did author apply in the formula given?

Response: Thank you very much for raising this point. We have revised the sentences for more explanation as follows. 

“To calculate the productivity loss of T21 patients for 35 years, we calculated expected income each year with the increasing rate of 4%, which was obtained from an annual income growth rate during 2012-2022 in Thailand [24]. Based on the recommendation from the Thai HTA guidelines, since cost values are different in different time periods, future values of total expected income or productivity loss during 35 years (FV) should be adjusted to their present values (PV) using an annual discount rate of 3% based on this formula: PV = FV x [1/(1+r)n, where PV = present value, FV = future value, r = discount rate, and n = each year in the future [19].” (Indirect cost, line 169-176, page 8-9)

7. Table 2: what is the unit of annual direct medical cost for OPD and IPD? Is it per visit, per admission, or per patient? Table 1 shows number of visits, and LOS. Authors should classify them by CHD.

Response: Thank you very much for pointing this out. As we have added “Table 1” to demonstrate parameters used for cost calculation (page 9). Therefore, Table 2 is now changed to Table 3 which shows the unit of annual direct medical cost for OPD and IPD per patient. Please see the revised title of Table 3 (page 12). In addition, Table 7 illustrates total annual direct medical costs incurred with an ‘ever’ diagnosis of CHD for trisomy 21 patients as suggested (page 17). 

8. Drug and medical device are the main component of costs. Do they include drug costs for co-morbidities?

Response: Yes, drug costs for co-morbidities were included, as cost data of drug and medical devices for all treatment were analyzed in our study. We have revised the sentence to make it clearer as follows. 

“Direct medical costs included service fee, drug and medical devices for trisomy treatment and co-morbidities, laboratory diagnosis, radiology examination, rehabilitation, operations, and other services such as dental care, psychology, and blood transfusion.” (Study design, line 95-98, page 5).

9. Table 6 displays the annual average cost. Indirect costs of each trisomy were equal. Why? Furthermore, indirect cost of patient was equal to that of care giver? Patients were children, but care givers were generally their parents.

Response: Thank you very much for raising this point. After carefully checking, we have re-analyzed and found that the indirect costs of T13 and T18 were equal, but those of T21 were different. Thus, we have changed the results in Table 8 (page 18). 

For T13 and T18 patients, the indirect costs (productivity loss) were estimated for only caregivers who took care of child patients by the average survival of most patients with T13 and T18 equal to one year multiplied by the Thai Gross Domestic Product (GDP) per capita per year (7,233.4 USD). On the other hand, for T21 patients, the indirect costs were calculated for both caregivers and patients and estimated by the average expected working year of T21 patients (35 years) i.e., average survival years (50 years) minus working age adults at 15 years multiplied by the Thai GDP per capita per year (7,233.4 USD). Even though caregivers were generally their parents who took care of their child patients, they would lose the opportunity to work for gaining income or have productivity loss. Therefore, in this study we applied a human capital approach to calculate the indirect costs or productivity loss of caregivers. 

We have added more explanation on how to calculate indirect costs under “Indirect cost” sub-heading (line 152-176, page 8-9). 

10. According to Table 1, age range of patient with trisomy 13 and trisomy 18 was 0-14 years. Table 6. Why is lifetime cost equal to annual total cost for Trisomy 13 and Trisomy 18? Patients can live longer than a year. 

Response: Thank you very much for your raising this point. We did not use age range of patients with T13 and T18 in our study because it might not reflect the true survival of most patients with T13 and T18. According to published studies on survival of children with T13 and T18, median survival was only 5 and 8 days, respectively. Around 11-17% of T13 newborns and 13% of T18 newborns could survive beyond their first year of life (1). In addition, 10-16% of T13 patients and 10-12% of T18 patients could reach 5 years and only 10% could survive at 10 years (2). There are rare reported cases of these patients surviving into late childhood (3,4). Overall, approximately 90% of these patients would die within one year. Therefore, we applied the survival of one year to calculate the productivity loss of caregivers for these child patients in this study. 

References:

1. Meyer RE, Liu G, Gilboa SM, Ethen MK, Aylsworth AS, Powell CM, et al. Survival of children with trisomy 13 and trisomy 18: A multi-state population-based study. Am J Med Genet A. 2016;170A(4):825-37. 

2. Glinianaia SV, Rankin J, Tan J, Loane M, Garne E, Cavero-Carbonell C, et al. Ten-year survival of children with trisomy 13 or trisomy 18: a multi-registry European cohort study. Arch Dis Child. 2023;108(6):461-7.

3. Goff RD, Soares BP. Neuroradiological findings of trisomy 13 in a rare long-term survivor. Neuroradiol J. 2018;31(4):412-4.

4. Khan F, Jafri I. Characterization of a 16-Year-Old Long-Time Survivor of Edwards Syndrome. Cureus. 2021;13(5):e15205.

In addition, how did author calculate lifetime cost? The detail should be added.

Response: We have added more details on lifetime cost calculation at the last paragraph of “Indirect cost” sub-heading (Indirect cost, line 167-176, page 8-9) 

11. Authors might display some results using pictures instead of tables.

Response: Thank you very much for your helpful suggestion. We have changed a table of percentage of direct medical costs for individuals with trisomy 13, 18, 21 to figure 1. 

Reviewer #2: 

I thank the editor for the opportunity to review this manuscript. I also commend the authors for undertaking this important work to delineate the cost burden faced by trisomy patients and their caregivers. Overall, the paper is sound and clear with the necessary analyses done. However, I do have concerns with the way the results have been presented in the tables and the discussion section. The discussion section needs to be more concise and clearly link the findings back to the existing evidence. While the authors have attempted this, it seems as though the authors are merely listing other studies rather than to make meaningful linkages back to the findings and how it adds value to the topic/field. I have provided some brief comments below and have added more detailed feedback and edits to the attached document.

Response: Thank you very much for taking your time to review our manuscript. We feel that our manuscript has been much further improved according to your great suggestions. 

1. It seems a little confusing that the first sentence says "cross-sectional study" but the second sentence says it is a "retrospective" study. I would think the authors are saying that the electronic database was retrospectively screened to identify patients with T13, T18, T21 and calculate the direct medical costs for a one-year period. If this is right, please amend as suggested.

Response: Thank you very much for your useful suggestion. To avoid the confusion, we deleted "cross-sectional study" and revise the sentence as suggested. 

“The inpatient and outpatient care data of patients with T13, T18, and T21 for direct medical cost calculation in 2016 were retrospectively retrieved from the Siriraj Informatics and Data Innovation Center (SiData+), Faculty of Medicine, Siriraj Hospital.” (Study design, line 112-116, page 6) 

2. Please elaborate exactly the data fields that were collected to aid in the cost calculation. Maybe including a table of variables that were extracted from the electronic database would be good (unless it is sensitive) to include here so that readers could know what parameters were used to calculate the direct medical costs. You could also include in the table what was used to calculate the indirect costs as well. 

Response: We have added “Table 1” showing parameters used to calculate direct medical, direct non-medical, and indirect costs (page 9). 

3. I'm aware 15 is the legal age for consent in Thailand. However, may I clarify that children are treated as adults at 15 years old? May I also just clarify the rationale for setting 15 years old as the cut-off and not 18 years, which is the effective age of consent please?

Response: Yes, the legal age for consent is Thailand is 15 years old. We have already added this sentence at “Direct medical and non-medical costs” sub-heading (line 136-139, page 7). 

“Even though the effective age of consent in Thailand is 18 years old, we applied 15 years old as the cut-off, since hospitals in Thailand routinely provide care for pediatric patients at age ranging from newborn to 15 years old.”

4. How was patient data handled? Were they all anonymized data? Suggest to include some information about how the extracted data was handled - i.e. were there any personal identifiers, how was the data stored, how long was it stored, was it password-protected, who had access etc. under the methods section. This is important in ensuring responsible use of patient data in studies.

Response: Thank you very much for your great suggestion. We have already added more explanation how patient data were handled in the method section as suggested.

“Patient data from all wards and departments were obtained except trauma department which might not be related to trisomy patients’ co-morbidities. For personal identifiers, coding was used instead of patient’s name and personal data. All raw data were recorded in the principal investigator’s computer with protection code. Principal investigator was the only person who could get access to the data. The data would be stored in the computer until the study has been published. The ethics approval was granted by the Siriraj Institutional Review Board (SIRB) (MU-MOU COA 657/2021), and the requirement for informed consent was waived by the SIRB committee.” (Study population, line 123-130, page 6-7)

6. For all tables, suggest to break them into 2 separate sub-tables for outpatient and inpatient care so that the tables do not look too packed.

Response: Thank you very much for your great suggestion. We have separated a table of total annual direct medical costs for individuals with Trisomy 13, 18, 21 classified by type of costs (USD) for outpatient (Table 4, page 14) and inpatient care (Table 5, page 15) as suggested.

7. Please standardize the number of decimal places/significant figures for p-values. For instance, you can leave it as 2 sig figures. For p-values like this, you can indicate as p <0.01. 

Response: Thank you very much. We have changed p-values to “<0.01”. (Table 6, page 16 and Table 17, page 17))

Again, what is the rationale for including gender here? Not really necessary to put in the breakdown of male and female since it doesn't contribute to the cost you are looking at here.

Response: Thank you very much. We have deleted the breakdown of males and females. (Table 17, page 17)) 

8. Does patient include both outpatients and inpatients? The number does not seem to tally. For example, T21 - 241 outpatients and 124 inpatients = 365 patients in total. For T13 and T18, there is no differentiation between inpatient and outpatients. Is there a reason for this? Suggest to provide the figures separately for inpatients and outpatients for T13 and T18 - what is the number of inpatients with T13 and outpatients with T13 and vice versa for T18 for clarity.

Response: We have checked the figure and the total number of patients was 377 patients. As T13 and T18 patients received both outpatient and inpatient cares, the total number of outpatient and inpatient visits was higher than the number of patients. We have added the sentence below under “Demographic characteristics of patients” sub-heading as suggested. 

“Total of 377 patients with T13 (5 patients), T18 (7 patients), and T21 (365 patients) were included in our analysis. All patients with T13 (5 patients) and T18 (7 patients) received both outpatient and inpatient care services, whereas 241 patients with T21 had outpatient visits and 124 received inpatient care.” (line 194-198, page 10)

9. The table of Total annual direct medical costs with Trisomy 13, Trisomy 18, Trisomy 21 (USD) could be better presented - is there a need to show the no. of males and females here? Maybe this could be reflected in the demographics table instead. For this table, it would be good to just focus on outpatient/inpatient and T13, T18, T2. Suggest to maybe switch the columns and rows for the table to be better presented.

Response: Thank you very much for your very helpful suggestion. There is no need to show number of males and females, therefore we deleted it from Table 3 (page 12). We have switched the columns and rows for the table to be better presented as suggested.

Summary of comments and suggestions from track change on the manuscript

1. “In order to illustrate and generalize in Thailand, it is necessary to employ up-to-date, locally-relevant lifetime costs in the cost-effectiveness analysis of prenatal screening tests.” This does not sound so smooth. What are you trying to generalize?

Response: Thank you very much. We have already revised the sentence to make it more understandable as follows.

“Our study could provide up-to-date and locally-relevant lifetime costs of patients with T13, T18, and T21 which are necessary to be applied in the cost-effectiveness analysis of prenatal screening tests.” (Introduction, line 87-89, page 5)

2. Can the authors elaborate briefly how the costs were calculated please? Did the family of the patients provide any cost-related information to this study?

Response: Yes, we have already elaborated more how direct non-medical costs were calculated as suggested. The family of the patients did not provide any cost-related information to this study.

“Direct non-medical costs i.e., transportation and meal expenses incurred by patients and their families during outpatient and inpatient visits were calculated by the average number of outpatient care visits and length of stay (LOS) per case analyzed from the hospital electronic database multiplied with the unit costs of transportation per one round trip and three meals obtained from the standard cost list for health technology assessment (HTA).” (Study design, line 100-105, page 5)

3. This answers my previous comment on what constitutes the direct costs but maybe this could be made more clearer in a table rather than mentioning this in the paragraph. You could just say "direct costs were calculated using the parameters as indicated in Table X". You could also include in the table what was used to calculate the indirect costs as well. Just a suggestion for the authors.

Response: We have added “Table 1” showing parameters used to calculate direct medical, direct non-medical, and indirect costs (page 9).

4. Was only transport and meals considered under direct non-medical costs? Were there other variables included? If so, you can provide all these variables in a table as suggested above.

Response: Yes, only transport and meals was considered under direct-non medical costs. We have added “Table 1” showing parameters used to calculate direct medical, direct non-medical, and indirect costs (page 9).

5. “were discounted at a rate of 3% per year.” Please clarify what this means.

Response: Thank you very much for raising this point. We have revised the sentences for more explanation as follows. 

“To calculate the productivity loss of T21 patients for 35 years, we calculated expected income each year with the increasing rate of 4%, which was obtained from an annual income growth rate during 2012-2022 in Thailand [24]. Based on the recommendation from the Thai HTA guidelines, since cost values are different in different time periods, future values of total expected income or productivity loss during 35 years (FV) should be adjusted to their present values (PV) using an annual discount rate of 3% based on this formula: PV = FV x [1/(1+r)n, where PV = present value, FV = future value, r = discount rate, and n = each year in the future [19].” (Indirect cost, line 169-176, page 8-9)

6. Does this figure (371 patients) include both outpatients and inpatients? The number does not seem to tally. For example, T21 - 241 outpatients and 124 inpatients = 365 patients in total. For T13 and T18, there is no differentiation between inpatient and outpatients. Is there a reason for this? Suggest to provide the figures separately for inpatients and outpatients for T13 and T18 - what is the number of inpatients with T13 and outpatients with T13 and vice versa for T18 for clarity.

Response: We have checked the figure and the total number of patients was 377 patients. As T13 and T18 patients received both outpatient and inpatient cares, the total number of outpatient and inpatient visits was higher than the number of patients. We have put the number of inpatients with T13 and outpatients with T13 and vice versa for T18 in Table 2 as suggested (page 11). 

7. Table 1 add N

Response: N has been added in Table 2 which was Table 1 previously (page 11). 

8. “Cost components” Better to include a header here since you are zooming in on the components that made up the total direct medical costs

Response: Thank you very much. We have changed a header to “Total direct medical cost”. (line 210, page 13)

9. “Annual Direct medical cost (USD)” Standardize your headers - either capitalize all the first letters or only the first letter

Response: Thank you very much. We have already corrected this (page 18). 

10. At the 2nd paragraph of discussion part, what point is this paragraph trying to make? That the mean annual direct medical costs for T21 is lower? The comparison of costs between the different countries could be better phrased

Response: Yes, we would like to explain that mean annual direct medical costs for T21 patients was lower than other countries. We have revised this paragraph by connecting with the first paragraph, as they are the same points.

“Similarly, mean annual direct medical costs (2,740 USD) in this study was much lower compared to the studies in Australia [24], the US [11, 25]. and Korean [26]. This can be explained that direct healthcare costs incurred in Thai hospitals are much lower compared to those in developed countries.” (Discussion, line 267-270, page 19)

11. “Nevertheless, compared to lifetime economic burden of T21 patients in Asia countries like China (55,000 USD) [30] and Thailand (72,000 USD) [15], our lifetime economic burden of T21 patients was higher.” Maybe better to provide reasons why?

Response: Thank you very much. We have added the reasons as suggested.

“Nevertheless, compared to lifetime economic burden of T21 patients in Asia countries like China (55,000 USD) [33] and Thailand (72,000 USD) [15], our lifetime economic burden of T21 patients was higher for the reason that our direct medical costs data were collected from the largest teaching and tertiary hospital where leading-edge and high cost technology are available to care for these patients and we assumed that T21 patients would receive treatment and have productivity loss for 35 years. Thus, this may overestimate the lifetime cost of T21 patients in our study.” (Discussion, line 312-318, page 21)

12. Last paragraph of discussion part. How does this study contribute to the existing gap? 

Response: Thank you very much for great suggestions. We have added the sentences as suggested.

“To the best of our knowledge, this is the first study to demonstrate both direct medical, direct non-medical and indirect costs of patients with T13, T18, and T21 in Thailand which could fill in the gap with the latest lifetime costs of these patients specifically in local Thai context. This information would be necessary to be applied in the cost-effectiveness analysis of prenatal screening tests which can be used for policy decision making whether the tests should be included in the benefit package of the Universal Health Coverage. In addition, the results from this study can help support the Thai government to implement the prenatal screening test policy to all pregnant women in the near future.” (Discussion, line 319-326, page 21)

13. How are the findings important in pushing for more pre-natal screening etc? Link the findings back to address the current gaps.

Response: Thank you very much for great suggestions. We have added the sentences as suggested.

 “To the best of our knowledge, this is the first study to demonstrate both direct medical, direct non-medical and indirect costs of patients with T13, T18, and T21 in Thailand which could fill in the gap with the latest lifetime costs of these patients specifically in local Thai context. This information would be necessary to be applied in the cost-effectiveness analysis of prenatal screening tests which can be used for policy decision making whether the tests should be included in the benefit package of the Universal Health Coverage. In addition, the results from this study can help support the Thai government to implement the prenatal screening test policy to all pregnant women in the near future.” (Discussion, line 319-326, page 21)

14. Conclusion, Repetition of findings

Response: Thank you very much. We have revised conclusion as suggested. 

“Our study demonstrates direct medical, direct non-medical, indirect costs as well as the lifetime costs of Thai patients with T13, T18, and T21 in a tertiary hospital based on a societal perspective. The results confirm that there is a significant economic burden of patients with T13, T18, and T21 in Thailand. The findings from this study could provide up-to-date direct and indirect cost information for trisomy patients especially in Thai context which would be the best available evidence for conducting economic evaluation of prenatal screening for policy makers 339-345, page 22)

---

## [Decision Letter · Decision Letter 1]

17 Jul 2023

PONE-D-23-05563R1Economic Cost of Patients with Trisomy 13, 18, 21 in a Tertiary Hospital in ThailandPLOS ONE

Dear Dr. Chaikledkaew,

Thank you for submitting your manuscript to PLOS ONE. After careful consideration, we feel that it has merit but does not fully meet PLOS ONE’s publication criteria as it currently stands. Therefore, we invite you to submit a revised version of the manuscript that addresses the points raised during the review process.

 We have invited the same reviewers and they have some additional comments which we invite you to address.

We look forward to receiving your revised manuscript.

Kind regards,

Yee Gary Ang, MBBS MPH

Academic Editor

PLOS ONE

Journal Requirements:

Reviewers' comments:

Reviewer's Responses to Questions

**Comments to the Author**

1. If the authors have adequately addressed your comments raised in a previous round of review and you feel that this manuscript is now acceptable for publication, you may indicate that here to bypass the “Comments to the Author” section, enter your conflict of interest statement in the “Confidential to Editor” section, and submit your "Accept" recommendation.

Reviewer #1: (No Response)

Reviewer #2: (No Response)

2. Is the manuscript technically sound, and do the data support the conclusions?

Reviewer #1: Yes

Reviewer #2: Yes

3. Has the statistical analysis been performed appropriately and rigorously? 

Reviewer #1: Yes

Reviewer #2: Yes

4. Have the authors made all data underlying the findings in their manuscript fully available?

Reviewer #1: (No Response)

Reviewer #2: Yes

5. Is the manuscript presented in an intelligible fashion and written in standard English?

Reviewer #1: Yes

Reviewer #2: Yes

6. Review Comments to the Author

Reviewer #1: Overall, authors have addressed almost all comments. However, some issues need to be clarified.

Page 8 line 163-166, line 169-171: The estimation of productivity loss was 35 years multiplied by Thai GDP (7233 USD). Is it for caregivers and patients or only caregivers? Line 164 specified both caregivers and patients. Based on the above estimation, why did productivity loss of caregivers for Trisomy 21 was equal to 9255 USD from Table 8? In addition, productivity loss of patients with trisomy 21 was higher than caregivers who are healthy working people (Table 8)?

Reviewer #2: Thank you for addressing the reviewers' comments. The revised manuscript is much improved. However, there are still grammer /sentence structure issues that need to be addressed. Please refer to the attached file for suggested edits. Overall, the paper does contrbute significant findings about the economic burden of genetic mutations and would be useful to inform policy decisions. My suggestion would be to review the paper thoroughly to ensure that there are no issues with standard English/grammatical errors etc. Apart from this, I feel that the paper is of pubishable quality.

7. PLOS authors have the option to publish the peer review history of their article (what does this mean?). If published, this will include your full peer review and any attached files.

Reviewer #1: No

Reviewer #2: No

---

## [Author Response · Author response to Decision Letter 1]

23 Aug 2023

Response to Journal Requirements:

Response: Thank you very much for your suggestion. We have already reviewed the references to ensure that it is complete and correct.

Response to Reviewer:

Reviewer #1: 

Overall, authors have addressed almost all comments. However, some issues need to be clarified.

Response: We would like to thank the reviewers for their helpful comments and suggestions. We feel that the revised paper is much further improved as a consequence of their inputs. Please kindly see our responses to below issues. 

Page 8 line 163-166, line 169-171: The estimation of productivity loss was 35 years multiplied by Thai GDP (7233 USD). Is it for caregivers and patients or only caregivers? 

Response: The estimation of productivity loss was 35 years multiplied by Thai GDP and this is for T21 patients. We have added more clarification as suggested. 

(Indirect cost, line 161-171, page 8)

“In this study, we calculated the productivity loss of both caregivers and T21 patients with the average expected survival of 50 years [2], while we considered the productivity loss of only caregivers for T13 and T18 patients who had average survival of only one year according to published studies [1, 21-22]. Therefore, productivity loss of caregivers for T21 patients was estimated by the average expected survival of 50 years, since caregivers have to leave their jobs for caring multiplied by the Thai GDP per capita per year (7,233.4 USD) [23], whereas that of patients with T21 was estimated by the average expected working year of T21 patients (35 years) i.e., average survival years (50 years) minus working age adults at 15 years multiplied by the Thai GDP per capita per year. The summation of productivity loss was divided by 35 years for T21 patients and 50 years for their caregivers to generate an annual productivity loss.”

Line 164 specified both caregivers and patients. Based on the above estimation, why did productivity loss of caregivers for Trisomy 21 was equal to 9255 USD from Table 8? In addition, productivity loss of patients with trisomy 21 was higher than caregivers who are healthy working people (Table 8)?

Response: Thank you very much for pointing this out. The productivity loss was calculated for both T21 patients and their caregivers. Table 8 presents the annual productivity loss of trisomy patients and their caregivers. The annual productivity loss of caregivers for T21 (9255 USD per year) was calculated by the summation of productivity loss of T21 patients (354,819 USD) divided by 35 years. On the other hand, the productivity loss of T21 patients (9,255 USD per year) was calculated by the summation of productivity loss of caregivers (462,733 USD) divided by 50 years. It is noted that the total productivity loss of caregivers for T21 was higher than that of T21 patients. We have carefully checked our calculation and added the corrected numbers in Table 8 (page 19-20). We have already added more clarification as follows.

(Indirect cost, line 161-181, page 8-9)

“In this study, we calculated the productivity loss of both caregivers and T21 patients with the average expected survival of 50 years [2], while we considered the productivity loss of only caregivers for T13 and T18 patients who had average survival of only one year according to published studies [1, 21-22]. Therefore, productivity loss of caregivers for T21 patients was estimated by the average expected survival of 50 years, since caregivers have to leave their jobs for caring multiplied by the Thai GDP per capita per year (7,233.4 USD) [23], whereas that of patients with T21 was estimated by the average expected working year of T21 patients (35 years) i.e., average survival years (50 years) minus working age adults at 15 years multiplied by the Thai GDP per capita per year. The summation of productivity loss was divided by 35 years for T21 patients and 50 years for their caregivers to generate an annual productivity loss.

To calculate the productivity loss of T21 patients for 35 years and caregivers for 50 years, we calculated expected income each year with the increasing rate of 4%, which was obtained from an annual income growth rate during 2012-2022 in Thailand [24]. Based on the recommendation from the Thai HTA guidelines, since cost values are different in different time periods, future values of total expected income or productivity loss during 35 years (FV) should be adjusted to their present values (PV) using an annual discount rate of 3% based on this formula: PV = FV x [1/(1+r)n, where PV = present value, FV = future value, r = discount rate, and n = each year in the future [19]. In addition, the lifetime costs were calculated by the summation of direct medical costs, direct non-medical costs and indirect costs of caregivers during one year for T13 and T18 patients, 35 years for T21 patients and 50 years for T21 caregivers.”

Reviewer #2: 

We would like to thank the reviewers for their helpful comments and suggestions. We feel that the revised paper is much further improved as a consequence of their inputs. Please kindly see our responses to below issues. 

1. Do you mean assigning pseudonyms here instead of using the names of patients? If 

so, please specify it as such as "coding" here is ambiguous

Response: Thank you very much for your great suggestion. We have changed “coding” to “assigning pseudonyms” as follows. 

(Study population, line 126, page 6)

“For personal identifiers, assigning pseudonyms were used instead of patient’s name and personal data.”

2. Was it an encrypted device? Please state as "in PI's encrypted computer with 

password protection" if so.

Response: Thank you very much for your great suggestion. We have changed to "in PI's encrypted computer with password protection" as suggested.

(Study population, line 127-128, page 6)

“All raw data were stored in the principal investigator’s encrypted computer with password protection.”

3. What about the research team members? Were they allowed access to the data at all? 

If they were given access to the data for analysis, please state it clearly here.

Response: Thank you very much for your great suggestion. The research team members were not allowed access to the data. We have added the below sentence to make it clearly. 

(Study population, line 128-129, page 6-7)

 “Principal investigator was the only person who could get access and analyze the raw data and other researchers could see only summary tables or charts.” 

4. Table 2 Shouldn't this be Adult >=15 instead?

Response: Thank you very much for your great suggestion. We have changed to “Adult >=15” in Table 2 (page 12). 

5. I'm not sure what this reference to the Taiwan's national health insurance means here. 

Are the authors trying to say it is compared to a "study conducted in a hospital in Taiwan?"

Response: Thank you very much for your great suggestion. Yes, we want to state “a study conducted in Taiwan”. We have changed the sentence as suggested. 

(Discussion, line 282-283, page 21)

 “However, our study showed similar average direct medical costs among T21 adults to the study conducted in a hospital in Taiwan (5,006 USD) [30].”

6. Would it be better to refer to patients who are children as "pediatric" patients? 

Response: Thank you for your great suggestion. Yes, we agreed with your suggestion. When we refer to patients who are children, we have changed “children” to “pediatric” patients throughout our manuscript. 

7. Please rephrase it to "our study found that caregivers were more likely to be female" 

rather than the way it is currently phrased.

Response: Thank you for your great suggestion. We have revised the sentence as suggested. 

(Discussion, line 310-311, page 23)

“Furthermore, our study found that caregivers were more likely to be female.”

---

## [Decision Letter · Decision Letter 2]

11 Sep 2023

Economic Cost of Patients with Trisomy 13, 18, and 21 in a Tertiary Hospital in Thailand

PONE-D-23-05563R2

Dear Dr. Chaikledkaew,

We’re pleased to inform you that your manuscript has been judged scientifically suitable for publication and will be formally accepted for publication once it meets all outstanding technical requirements.

Kind regards,

Yee Gary Ang, MBBS MPH

Academic Editor

PLOS ONE

Reviewers' comments:

Reviewer's Responses to Questions

**Comments to the Author**

1. If the authors have adequately addressed your comments raised in a previous round of review and you feel that this manuscript is now acceptable for publication, you may indicate that here to bypass the “Comments to the Author” section, enter your conflict of interest statement in the “Confidential to Editor” section, and submit your "Accept" recommendation.

Reviewer #1: All comments have been addressed

Reviewer #2: All comments have been addressed

2. Is the manuscript technically sound, and do the data support the conclusions?

Reviewer #1: Yes

Reviewer #2: Yes

3. Has the statistical analysis been performed appropriately and rigorously? 

Reviewer #1: Yes

Reviewer #2: Yes

4. Have the authors made all data underlying the findings in their manuscript fully available?

Reviewer #1: Yes

Reviewer #2: Yes

5. Is the manuscript presented in an intelligible fashion and written in standard English?

Reviewer #1: Yes

Reviewer #2: Yes

6. Review Comments to the Author

Reviewer #1: Authors have addressed all comments. The revised version of the manuscript is more interesting than the previous version. There is no further comment.

Reviewer #2: I'm satisfied that the authors have addressed the feedback given from the previous rounds of review and the manuscript is now of good quality to be published.

7. PLOS authors have the option to publish the peer review history of their article (what does this mean?). If published, this will include your full peer review and any attached files.

Reviewer #1: No

Reviewer #2: No

---

## [Editor Report · Acceptance letter]

13 Sep 2023

PONE-D-23-05563R2 

Economic cost of patients with trisomy 13, 18, and 21 in a tertiary hospital in Thailand 

Dear Dr. Chaikledkaew:

I'm pleased to inform you that your manuscript has been deemed suitable for publication in PLOS ONE. Congratulations! Your manuscript is now with our production department. 

Kind regards, 

on behalf of

Dr. Yee Gary Ang 

Academic Editor

PLOS ONE